# Manufacturing and Analysis of High-Performance Refractory High-Entropy Alloy via Selective Laser Melting (SLM)

**DOI:** 10.3390/ma12050720

**Published:** 2019-03-01

**Authors:** Hang Zhang, Yizhen Zhao, Sheng Huang, Shuo Zhu, Fu Wang, Dichen Li

**Affiliations:** School of Mechanical Engineering, Xi’an Jiao Tong University, State Key Laboratory of Manufacturing System Engineering, Xi’an 710049, China; zhanghangmu@mail.xjtu.edu.cn (H.Z.); zyz8zyz@stu.xjtu.edu.cn (Y.Z.); ahhuangsheng@126.com (S.H.); zhushuoxjtu@163.com (S.Z.); dcli@mail.xjtu.edu.cn (D.L.)

**Keywords:** additive manufacturing, high-entropy alloy (HEA), selective laser melting (SLM), microstructure, superalloys

## Abstract

Refractory high-entropy alloys (HEAs) have excellent mechanical properties, which could make them the substitutes of some superalloys. However, the high melting point of refractory HEAs leads to processing problems when using traditional processing techniques. In this study, a single BCC solid solution of NbMoTaW alloy was formed by selective laser melting (SLM) with a linear energy density of up to 2.83 J/mm. The composition distribution was analyzed, and the element with a lower melting point and lower density showed a negative deviation (no more than 5%) of the molar ratio in the formed alloy. The HEA shows an excellent microstructure, microhardness, and corrosion resistance performance compared with traditional superalloys, making it a new substitute metal with great application prospects in aerospace and energy fields.

## 1. Introduction

“Superalloy” is the general name given to iron-based, nickel-based, and cobalt-based alloys, given their high endurance, creep, and fatigue strength at high temperature. They are widely used in aviation, aerospace, and automotive and chemical industries, serving in high-temperature (≤ 1100 °C) environments. However, for super high temperature conditions of up to 1500 °C and 3000 °C, traditional surperalloys are powerless. Instead, a new type of refractory high-entropy alloys (HEAs) have shown their advantages. High-entropy alloys or multi-element alloys are loosely defined as solid-solution alloys that contain more than five principal elements in equal or near equal atomic percent (at%) [1,2,3]. HEAs usually have unexpected properties, such as high strength, high hardness, excellent softening resistance, and wear and corrosion resistance [4,5,6,7,8,9]. According to the cocktail effects of HEAs, the alloy properties can be adjusted by composition changes and via alloying [4,5,10]. Combined with high-heat-resistant elements, HEAs will have excellent properties such as a high melting point, which meets the requirements of high-temperature load-bearing structures and thermal protection systems for aerospace applications. Therefore, HEAs can be seen as potential substitutes of surperalloys. Accordingly, Senkov and others [11,12] prepared a body-centered cubic (BCC) structure refractory HEA with near-equiatomic concentrations, using NbMoTaW, which was produced by vacuum arc-melting. The yield strength for this refractory HEA was 1058 MPa at room temperature, and 405 MPa at 1600 °C, which is higher than the melting point of most existing nickel-based superalloys. This refractory HEA may have promising applications in many fields, such as aerospace, oceaneering, and energy fields.

However, the ductility of the NbMoTaW HEA at room temperature is low. Furthermore, the full mixing of elements in the refractory HEAs is still a challenge. To achieve a homogeneous distribution of elements in the alloy, the buttons are usually re-melted four times and flipped for each melt by vacuum arc melting [11,12]. The low ductility at room temperature and high strength of refractory HEAs render them very difficult to process [13]. Meanwhile, the melting point is also too high to be manufactured by traditional thermal forming, such as investment casting. For the casting process, the highest melting temperature is only 1800 °C, which is far from enough for the NbMoTaW alloy in this study. Besides, although the arc melting can go up to the melting point, this method cannot form an arbitrary shape.

Selective laser melting (SLM) is a powder-bed-based layer manufacturing technique, which can directly fabricate metal parts according to three-dimensional (3D) computer-aided design (CAD) data by selectively melting successive layers of metal powders [9,13]. SLM is a complicated process, in which a high density of energy inputs at each laser pulse cycle and heat constantly dissipates in different ways. The heat transfer does not only occur between solid, liquid, and gas but also includes interactions between plasma, electrons, and photons. Figure 1 shows a simple summary of thermal behaviors during an SLM scanning process. In fact, there are several heat dissipation methods, including heat being removed by plasma emission, reflected light, radiation, phase transition, thermal convection, and conduction. Occasionally, the particle spatter also removes heat directly from the system.

Compared with conventional manufacturing, SLM has the capability of producing 3D parts of complex shapes without tools and molds. Thus, SLM has many advantages, particularly in the processing of refractory metal materials. During the process, only a very small volume of material is heated by laser during a short interaction time; therefore, the molten pool can reach several thousand degrees instantaneously and the SLM process has very high cooling rates, which benefits the formation of solid solutions in HEAs and still forms grains in the final stage. In fact, some studies [14,15,16,17,18,19] have been conducted to prepare HEA directly by laser. Zhang et al. [14] synthesized a refractory HEA coating with composition close to TiZrNbWMo on C45 steel by laser cladding. The Al_0.3_CoCrFeNi HEA was fabricated with direct laser fabrication by Joseph et al. [15], who showed that the materials exhibited significant tension/compression asymmetry in work hardening rate and ductility of the alloy. Kunce and others [16,17] synthesized two HEAs, ZrTiVCrFeNi and TiZrNbMoV, from a blend of elemental powders near the equimolar ratio using laser engineered net shaping (LENS), and the two alloys both exhibited a two-phase structure as well as hydrogen storage capacity.

However, the manufacturing of NbMoTaW HEA is still a serious problem for traditional hot working. There have been notably few studies regarding the preparation of refractory HEA by SLM. Therefore, in this work, we examined the miscibility and solidification of NbMoTaW refractory HEA synthesized via SLM and conducted elemental, microhardness, and corrosion resistance analysis of the samples.

## 2. Material and Methods

The NbMoTaW HEA was fabricated with mixed multi-element powders. The powders of Nb (99.9%), Mo (99.9%), Ta (99.9%), and W (99.9%) were weighed with nominal composition in equal atomic percent, respectively. The shapes of powder grains were spherical or near-spherical. The particles were mixed with a mass ratio *m*_w_:*m*_Ta_:*m*_Mo_:*m*_Nb_ of 184:181:96:93, and the powders were then mechanically milled in a KQM-X/B planetary ball mill for 2 h and dried at 45 °C for 8 h.

In the experiment, the SLM process was conducted using FORWEDO LM120 equipment with a *P*_L_ (Maximum power of laser) of 500 W fiber laser in a protective atmosphere of argon with an amount of oxygen under 20 ppm. The substrate was C45 steel plate with a size of 50 mm× 50 mm × 20 mm. The fabricated sample model was a cuboid with a size around 10 mm × 10 mm × 0.5 mm.

The main parameters of the SLM process are laser scanning velocity (*v*), hatch distance (*ds*), layer thickness (*dz*), laser power factor (*f*_p_), and laser spot diameter (*d*_L_). The melting point of NbMoTaW is higher than that of most common metal alloys. Therefore, a proper higher energy input with a combination of laser power and scanning velocity is important for melting the mixed powder completely and shaping the HEA sufficiently. The original parameters were chosen as follows: *ds* was 0.1 mm, *dz* 0.1 mm, *f*_p_ 0.8, and *v* 250 mm/s, with S-cross scanning in *x* and *y* directions. Higher energy density is needed for fabricating NbMoTaW HEA parts, so there was an additional remelting process, which had the same parameters as the original process but underwent remelting twice for each layer, which means that there was 3 times the laser energy input. Therefore, the original process with a linear energy density of 0.943 J/mm, according to Equation (1), and the scanning strategy is illustrated in Figure 1.
(1)$$\rho_{Q}=(n+1)P_{L}f_{P}A_{mix}/v$$
where, $\rho_{Q}$ is the linear energy density, *n* is the number of remelting times, and *A_mix_* is the laser absorption rate of the mixed powder.

The samples for composition testing, named S1, S2, and S3, were fabricated by remelting processes. Subsequently, the samples were cut and their surfaces were analyzed further. The morphology of the microstructures on the specimen surfaces was observed in a HITACHI SU-8010 (Hitachi, Japan, Tokyo) scanning electron microscope (SEM), and the elemental analysis experiments were carried out by Energy Dispersive Spectrometer (EDS). The crystal structure was identified by an X-ray diffractometer D8 (XRD,) advanced with a Cu target (λ = 1.54 Å). The particle size was analyzed with the laser diffraction particle size analyzer Mastersizer 2000 (Malvern, UK), and the average particle sizes were investigated. 

The absorption rate (*A*) of the powder was indirectly measured. Subsequently, *A* of the powder to the fiber laser was investigated by testing the reflectance (*R*) of the elements. It can be assumed that transmission of the laser does not occur in metal powders, and the relationship between the absorption and the reflectance of metal is *A* = 1−*R* [20]. The reflectance of the powder was measured in a system by Avantes Co., The Netherlands. The grain and dendrite sizes were counted by the standard reference [21].

Besides, this material is expected to be mainly used in the field of marine engineering; thus, it needs to resist seawater corrosion, which was tested in this study. Therefore, for obtaining the potentiodynamic polarization curves to study the corrosion resistance ability of NbMoTaW HEA, three-electrode electrochemical corrosion tests were carried out for the HEA and 316 L stainless steel in the mass fraction of 3.5%wt NaCl solution. The patterns of HEA and 316 L steel, as the working electrode, were cut into a square of 6 mm × 6 mm. Then the surfaces of the patterns were encapsulated with non-conductive resin and only one surface was left. In addition, a calomel electrode was used as the reference electrode, and a platinum plate as the auxiliary electrode. The scanning voltage range was set from −0.2 V to +0.2 V.

## 3. Results and Discussion

### 3.1. Experimental Powders

The basic physical properties of the powders, such as shape and absorption of laser, are shown in Figure 2. The W, Ta, and Nb powders showed granular shapes, while the Mo powder showed a spherical shape. The order of the absorption rate of laser in the elements powder to the laser is *A*_Mo_ > *A*_W_ > *A*_Nb_ > *A*_Ta_, as shown in Figure 2, and the higher the absorption, the more energy the powder will receive at the same time. Thus, the average size of the powder is designed in the same corresponding order, *D*_Mo_ (94.9 um) > *D*_W_ (45.0 um) > *D*_Nb_ (21.5 um) > *D*_Ta_ (13.7 um). It is notable that the absorption of the mixed powder is higher than that of most powders except for Mo powder, suggesting that the mixing process helps to improve the absorption. The absorption of the Mo powder is higher than that of the mixed powder, since the Mo powder can trap the laser radiation easily in the hollow structure with gaps.

### 3.2. Phase and Composition

The X-ray powder diffraction (XRD) patterns of NbMoTaW HEA parts fabricated by SLM are exhibited in Figure 3a. The surface of the sample is smooth and has a distinctive metallic luster, which suggests relatively high density. The XRD results shown in Figure 3b, compared with Figure 3c, illustrate there is only one BCC structure solid solution without another metal phase in the alloy. Meanwhile, the experimental lattice parameter is 3.2034 Å, compared with the ‘theoretical’ crystal lattice parameter, *a*_mix_ = 3.2230 Å [12], which can be calculated with the following equation,
*a*_mix_ = ∑*c**_i_**a**_i_*(2)
where *c_i_* and *a_i_* are the atomic fraction and lattice parameter of element *i,* respectively. Besides, the diffraction peaks of (100), (200) and (211) are relatively concentrated, which means that the composition is uniform. The experimental results are in good agreement with those of the as-cast NbMoTaW HEA in References [11,12]. Therefore, it is obvious that the SLM process is an effective method to form NbMoTaW HEA.

After the phase confirmation of the parts, the element composition was examined, as shown in Figure 4a. The macroscopic composition is relatively uniform. The EDS analysis of the quinternary alloy shows that the four elements are uniformly distributed in macro scale as seen in Figure 3b. Besides, the composition distribution among micro grains is measured in Figure 4c. The amplitudes of composition fluctuations for Nb and Mo in micro scale are larger than those of Ta and W. It is obvious that the melting points of Nb and Mo are lower than those of Ta and W. During the cooling process, Ta and W first precipitate and distribute evenly in the base part, and Nb and Mo show segregation between inner grains and boundaries.

It can be seen that the elements in the samples are evenly distributed on the whole, but there is still segregation of components. Thus, three groups of sintering repeatability experiments were performed to verify the composition deviation of the samples. The composition distribution results and variance of NbMoTaW are showed in Table 1. From the table, although W and Mo approached the theoretical at% (atomic percent ) of 25% in all three samples, Ta had the largest average molar ratio (at%) with 28.21%, Nb had the lowest average ratio (21.50%), and their experimental fluctuations (0.99 and 0.16, respectively) in the three experiments are small. Thus, it is obvious that there is a certain degree of composition deviation in the patterns.

There are several factors leading to the composition deviation, such as melting point, liquid density, powder sides, and energy absorption of powders. According to Figure 5a–c, because the powder melting points, liquid densities, and at% have the same distribution trends, Mo and Nb, with lower melting points and densities, showed negative deviations of −0.16 at% and −3.5 at%, respectively, compared with the nominal composition of 25%. Therefore, in general, when mixed powder is heated up instantly by the laser, the powders with lower melting point are fused first and the liquid spreads on a larger surface area and absorbs more energy than the solid powders. Meanwhile, as the high-density energy continuously inputs, powders with high melting points are also fused, and there is a mixing process between different metal liquids. The liquid with lower density tends to float to the surface and the high-density liquid tends to drop, meaning that the lower-density liquid on the surface is exposed to the laser and receives more energy. In the end, these elements with lower melting points arrive at the boiling point earlier and evaporate more during the SLM process, leading to the negative composition deviation.

However, the final deviation results are actually combined with the factors of melting point, particle size, laser absorption, etc. Figure 5d shows the average particle size of four powders (black) and laser absorptivity (blue). It is obvious that the average particle size and laser absorptivity both have a similar variation trend and a certain degree of causal relationship. Meanwhile, combined Figure 5a,b, it is worth noting that although Mo and Nb have very similar melting points, the composition deviation between them is relatively huge, which is caused by particle size and laser absorption. Specifically, for Mo powder, although it has a higher laser absorption rate, the bigger powder size slows down the rate of melting, which causes lower mass loss and deviation compared with Nb.

In summary, it can be seen that the NbMoTaW HEA generated with the SLM process shows an approximate equimolar ratio distribution. Although there is a maximal 3.5% composition deviation, as a HEA, the constituent range from 5–35% is acceptable and the composition deviation does not influence the phase and microstructure. Actually, the NbMoTaW HEA formed by SLM can demonstrate three performance advantages.

### 3.3. Microstructure

Based on the SEM observation, on the XY plane, the grain boundaries of NbMoTaW HEA formed by SLM are clear in Figure 6a. The average grain size is 13.4 μm, which is much smaller compared with the 200 μm of as-cast NbMoTaW HEA grain size [12], as shown in Figure 6b. Besides, with a further amplification, in Figure 6c, many lamellar martensite structure dendrites appear on the top of the molten pools. The preferential orientations of the grains are in all directions and the second branches are interlaced with each other. The primary and secondary dendrite arm spacings are 6.59 μm and 1.68 μm on average, respectively, compared with the as-cast dendrite arm spacing of approximately 20 to 30 μm in Figure 6d [12].

Actually, under the rapid solidification condition of the laser processing, the grains and dendrites as a sub-structure grew and developed with an extremely high temperature gradient and cooling rate. The grains were extremely fine and the primary dendrite arm spacing was approximately half the size of a grain. This means that when the grain nuclei formed, their growth-driving force was large but the growth-time was limited. Thus, the dendritic branches were slender in shape, and the dendrite arm spacing was far less than that in other solidification processes such as casting.

The fine structure is thought to improve the mechanical properties of the final part. It has been reported that the as-cast NbMoTaW HEA is brittle with plastic strain ɛ_p_ = 0.2% [11], which seriously limits its potential applications. The well-known Hall–Petch (HP) law [22,23] illustrates the dependence of the grain size with the macroscopic plastic and the relationship between the brittle fracture stresses and polycrystal structure. Therefore, the plasticity, fatigue strength, and creep rate of the polycrystal BCC metal are improved when the grain size decreases [24]. Therefore, NbMoTaW alloy fabricated by SLM is a promising method for producing refractory HEA parts, although further study of mechanical properties is still necessary.

### 3.4. Microhardness

In this study, the microhardness (*H*_v_) of fabricated samples was measured on the XY plane, compared with the microhardness data of pure Nb, Mo, Ta, W, and four kinds of surperalloys [25,26]. The results are shown in Figure 7 and there are two points worth noting in the figure.

Firstly, the average microhardness value of the NbMoTaW HEA_s_ (sintering by SLM) sample is 826 Hv. However, the highest microhardness of Nb, Mo, Ta, and W is 410Hv and the mirohardness of surperalloys is only in the range of 310–437 Hv. Thus, it is obvious that the NbMoTaW HEA is much harder than superalloys and the pure elements it contains. Besides, according to the research of Senkov et al. [11], the NbMoTaW HEA does not produce abrupt hardness changes at high temperatures, which means that NbMoTaW HEA also has better hardness properties at high temperatures than superalloys.

Second, compared with the HEA_a_ (sintering by arc melting) sample (446 Hv) [12], the HEA_s_ sample also demonstrates a better hardness performance, which shows the advantage of grain refinement due to rapid cooling in the SLM process. It can also be seen that grain refinement can significantly improve the hardness of high-entropy alloys. Therefore, SLM can be an effective means to improve the hardness of high-entropy alloys.

### 3.5. Corrosion Resistance

According to the electrochemical theory, there are two parameters to characterize corrosiveness, self-corrosive potential (*E_corr_*) and self-corrosive current (*I_corr_*), among which self-corrosive potential is only a reference. The key parameter for judging the corrosiveness of metals is the free-corrosion current. A lower density (*I_corr_*) means a higher corrosion-resistant. In this study, the surface areas of two kinds of patterns are the same, so the free-corrosion current density can be replaced by the free-corrosion current. In Figure 8, the *I_corr_* of NbMoTaW HEA and 316L steel can be obtained by linear fitting: self-corrosion potential of HEA *E_corrH_* = −91.57 mV, free-corrosion current *I _corrH_* = 8.716 × 10^−11^ A; for 316 L steel, *E_corrS_* = −242.45 mV, *I_corrS_* = 8.815 × 10^−^9 A. As can be seen, the self-corrosion potential of HEA is more negative than that of 316L steel, which means the HEA has a stronger corrosion tendency. This is because the existence of a small amount of composition segregation in the alloy as mentioned before, which is readily forming corrosion couples and makes the alloy more “active.” However, the free-corrosion current of the HEA is reduced by two orders of magnitude compared with that of the 316 L steel. Therefore, in general, the NbMoTaW high-entropy alloy has better corrosion resistance. Actually, although a slight segregation of elements exists in the HEA, the elements Nb, Mo, and Ta contained in the alloy are all the easy passivation metals, which contributes to the excellent corrosion resistance in the 3.5 %wt NaCl solution.

Besides, in order to show the difference in corrosion resistance between the two materials more intuitively, the patterns of NbMoTaW and 316 L steel were also scanned at high voltage in the range of −1.5 to +2 V in 3.5%wt NaCl solution environment. In this solution, two kinds of patterns were corroded, but the degree of corrosion was quite different, as shown in Figure 9.

From Figure 9a, it is obviously that a strong corrosion happens in 316 L steel, with a corrosion depth of even up to 1 mm. However, in Figure 9b, for the NbMoTaW alloy, due to the effect of corrosion, only the crack of the sample has a certain extent of extension. At the same time, pitting corrosion also occurs on the smooth surface of the pattern. Therefore, from the extent of lost weight, the NbMoTaW HEA also shows stronger corrosion resistance than does 316 L steel.

## 4. Conclusions

In this paper, as a substitute material for superalloys, the NbMoTaW high-entropy superalloy could be formed by SLM. Besides, because of the melting point, liquid density, particle size, and energy absorption of powders, the element with the lowest melting point and density showed a negative deviation (no more than 5%) of molar ratio in the fabricated alloy. Fortunately, as a HEA, the constituent range of 5–35% is acceptable and the composition deviation does not influence the phase and microstructure. Actually, the NbMoTaW HEA formed by SLM can demonstrate three performance advantages.

The first is microstructure, as the microstructure analysis of the pattern revealed that the extremely fine grains and dendrites produced in the HEA sample by SLM had sizes of 13.4 μm and 6.59 μm, respectively, which are far smaller than those observed in as-cast samples. Second, the microhardness of NbMoTaW HEA_s_ can reach 826 Hv, which is much higher than the microhardness data of HEA_a_, pure Nb, Mo, Ta, W, and four kinds of surperalloys. This is caused by the effect of the grain refinement in the SLM process. Finally, regarding corrosion resistance, the free-corrosion current for NbMoTaW is *I_corrH_* = 8.716 × 10^−11^ A, and for 316 L steel, *I_corrS_* = 8.815 × 10^−9^ A; the free-corrosion current of HEA is reduced by two orders of magnitude compared with 316 L steel. Therefore, in general, the NbMoTaW high-entropy alloy has better corrosion resistance. 

## Figures and Tables

**Figure 1 materials-12-00720-f001:**
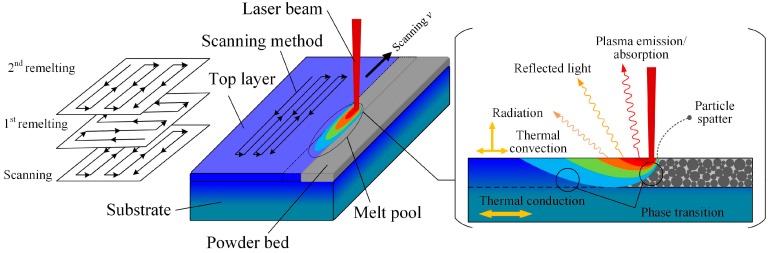
Heat dissipation process in selective laser melting (SLM).

**Figure 2 materials-12-00720-f002:**
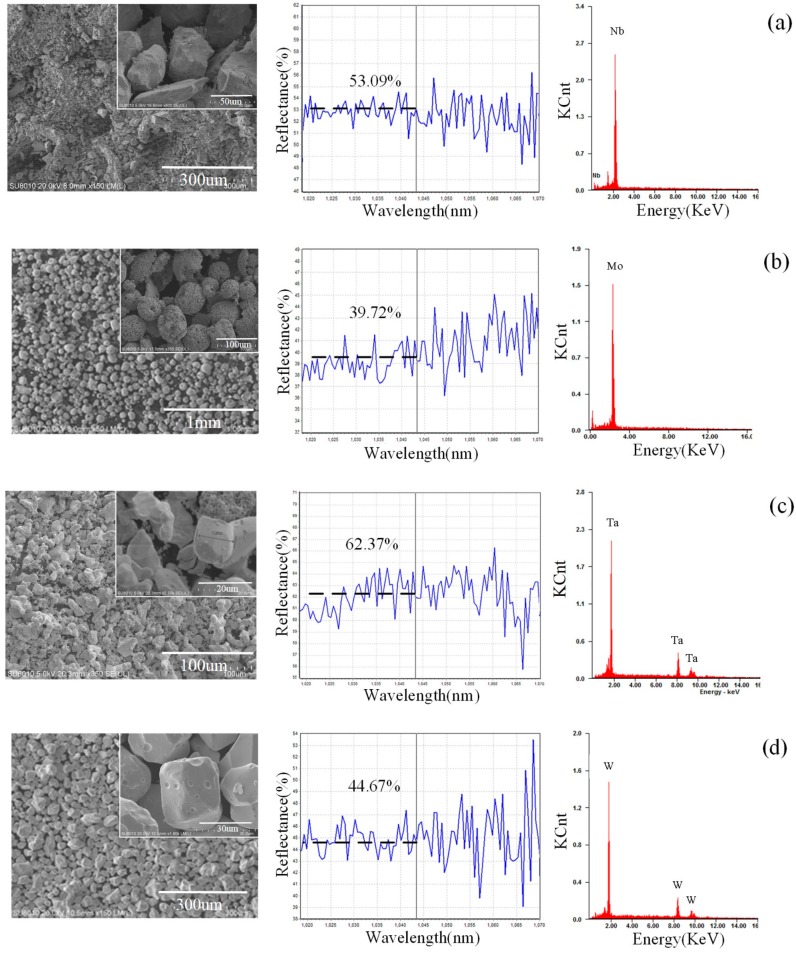
Basic physical properties of powders (left: shapes of powders; middle: laser absorption of powders; right: energy spectrums). (**a**) Nb, (**b**) Mo, (**c**) Ta, (**d**) W, and (**e**) mixed powders.

**Figure 3 materials-12-00720-f003:**
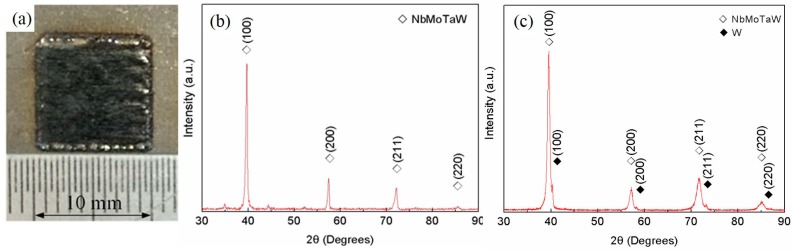
X-Ray powder diffraction (XRD) patterns of NbMoTaW HEA parts and results. (**a**) Patterns of NbMoTaW high-entropy alloy (HEA) parts fabricated by SLM. (**b**) XRD result with one alloy structure, (**c**) XRD result with alloy and a small amount of the other tungsten phase.

**Figure 4 materials-12-00720-f004:**
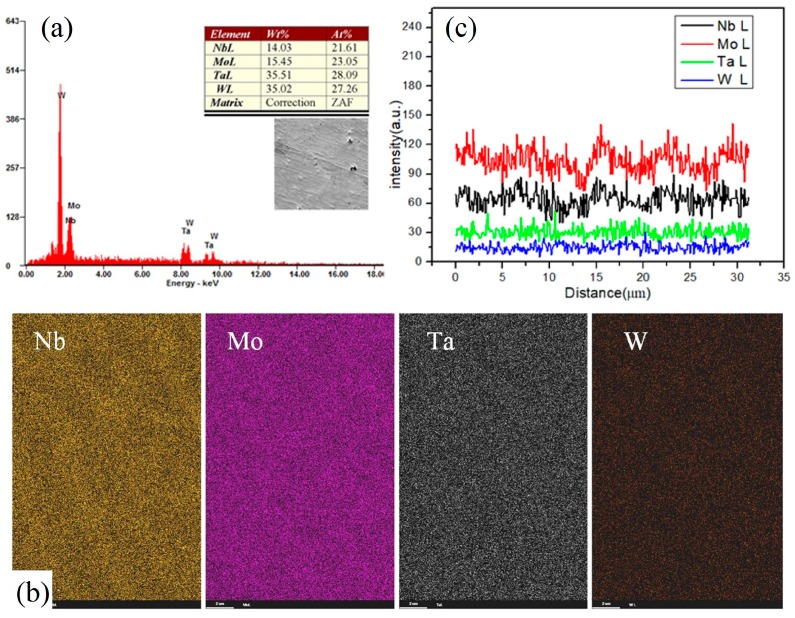
Results of composition analysis. (**a**) Composition distribution, (**b**) EDS analysis, (**c**) Composition distribution among micro grains.

**Figure 5 materials-12-00720-f005:**
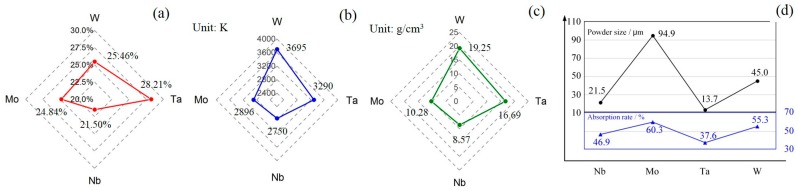
(**a**) Composition distribution results (mean value). (**b**) Melting point of Nb, Mo, Ta, W. (**c**) Liquid density of Nb, Mo, Ta, W. (**d**) Average particle size of four powders and laser absorptivity.

**Figure 6 materials-12-00720-f006:**
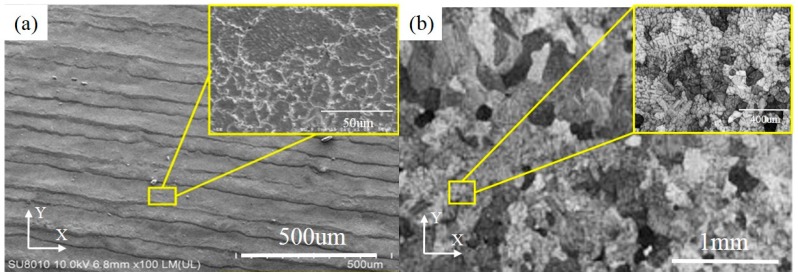
Different magnification figures of HEA pattern grain by scanning electron microscope (SEM) on XY plane. (**a**) Pattern grain figure in this study. (**b**) As-cast NbMoTaW HEA grain figure [12]. (**c**) Pattern grain figure with a further amplification in this study. (**d**) As-cast NbMoTaW HEA grain figure with a further amplification [12].

**Figure 7 materials-12-00720-f007:**
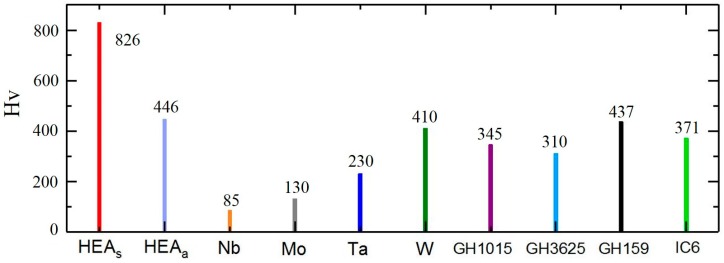
Microhardness (on XY plane) comparison for various pure metals and alloys [12,25,26].

**Figure 8 materials-12-00720-f008:**
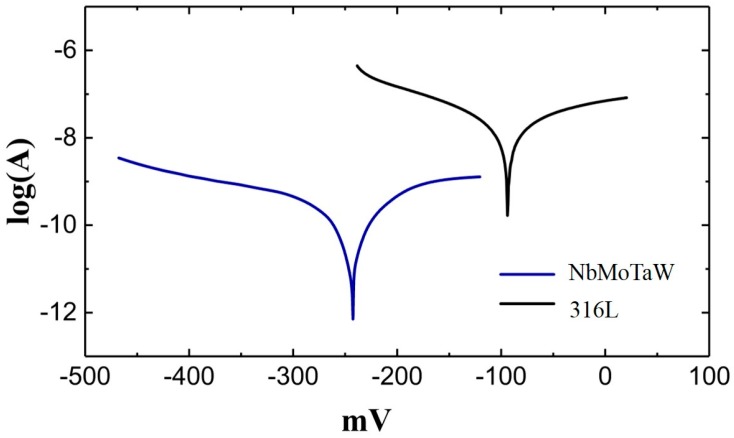
Polarization curve patterns of WMoNbTa HEA and 316 L.

**Figure 9 materials-12-00720-f009:**
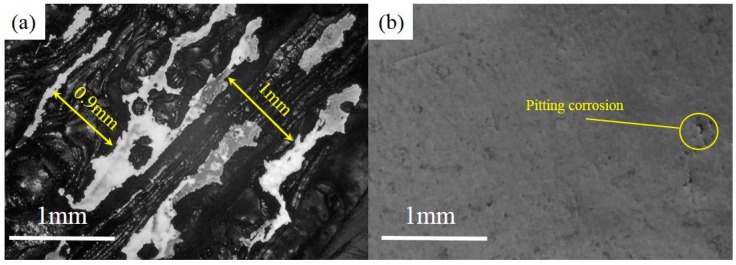
Corrosion of sample surface in high voltage range. (**a**) Sample of 316 L stainless steel. (**b**) Sample of NbMoTaW HEA.

**Table 1 materials-12-00720-t001:** Compositions of samples.

Element	S1 (at%)	S2 (at%)	S3 (at%)	Avg. (at%)	Variance
*W*	27.26	24.76	24.35	25.46	1.65
*Ta*	28.09	29.49	27.06	28.21	0.99
*Nb*	21.61	20.97	21.93	21.50	0.16
*Mo*	23.05	24.79	26.67	24.84	2.19

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
