# Peer review of "Manufacturing and Analysis of High-Performance Refractory High-Entropy Alloy via Selective Laser Melting (SLM)"

_materials, 2019, doi:10.3390/ma12050720_

Reviewer 1 Report

This investigation into SLM building of refractive HEAs can be highly useful in determining applicability of HEAs into engineering applications. While additive manufacturing of HEAs is picking up, the literature on refractory HEAs is more limited. There are some good insights into the complexity of multiple-element laser sintering involved in additive building HEAs, eg. Explanations of spatial segregation in Nb and Mo originating in their lower m.p.s and negative composition deviations are good. This study will help future endeavors in this area. But it needs some improvements and clarifications before this manuscript can become publishable. 

 1. The authors mention in the introduction section that SLM created HEA builds showed tension/compression asymmetry. Why did the authors not try to perform mechanical testing? Why did they limit to Hardness testing? Mechanical test data would make the paper stronger. I would recommend adding that data if possible. 

2. Why was a steel plate used as the substrate? Need to give reasoning eg. Conductivity similarity etc.? 

3. First paragraph of second section is a generic discussion of SLM process without any references to the current work at all. Hence this must be integrated into the introduction section. 

 4. Since the scanning pattern visited each spot three times, it cannot be said that the total input energy is three times the process’s linear energy density, 0.943 J/mm. So reporting 2.83 J/m in the abstract is wrong because it conveys the meaning that the incident energy per melt is that much. The end product from three times may help homogenization, but the microstructure will be very different from single scan with three times the energy. 

 5. It is mentioned that three samples wer eus.

6. It will be easier to read Figure 2 with elemental references embedded in the image. Also the third graph needs y-axis label to refer to the property being measured. 

7. I can see some smaller peaks than those labelled in Figure 3’s XRD pattern. All peaks need to be labeled since here the technique is being used to conclude the single phase formation. 

 8. Figure 5’s labels are a little hard to read. I would suggest increasing font-size. 

9. The microstructure in Figure 6(c) appears to be lamellar/widmanstatten (like in Ti64 high cooling rate) rather than dendritic/snow-like as described by the authors. I suggest the authors reconsider their interpretation or provide other images that better capture/represent the real microstructure. 

10. Since SLM structures are anisotropic in nature with columnar grain structure that is their trademark, the microstructure of the XZ and XY planes should be presented. 

11. Similarly, it is also important to know the orientation of sample in Hardness testing. It will be good to compare the hardness data from XY and XZ planes.

Author Response

Dear reviewer:

We are very grateful to your comments for the manuscript (ID materials-442714 " Manufacturing High-Performance Refractory High-entropy Alloy via Selective Laser Melting (SLM) "). Based on these comments and suggestions, we have made careful modifications on the original manuscript. All changes made to the text are highlighted in revision. Below you will find our point-by-point response to the reviewers’ comments/suggestions: 

1.     Specific Comments:

1.1 Reviewer Comments

The authors mention in the introduction section that SLM created HEA builds showed tension/compression asymmetry. Why did the authors not try to perform mechanical testing? Why did they limit to Hardness testing? Mechanical test data would make the paper stronger. I would recommend adding that data if possible.

Reply of the author: Thank you very much for your kind reminding. We are very sorry for our unclear report in mechanical testing. Due to the influence of process, we need to devote to more effort in printing thicker samples. Therefore, in this study, it is difficult for us to obtain the relevant data of mechanical properties, which we will focus on in the future research work.

1.2 Reviewer Comments

Why was a steel plate used as the substrate? Need to give reasoning eg. Conductivity similarity etc.?

Reply of the author: Thank you for your constructive and helpful suggestion. In this study, steel substrates are actually relatively good printing materials,compared with titanium alloy substrates. Of course, we will consider the effect of substrates materials on printing quality in future research.

1.3 Reviewer Comments

First paragraph of second section is a generic discussion of SLM process without any references to the current work at all. Hence this must be integrated into the introduction section.

Reply of the author: Thank you very much for your kind reminding. We have already modified in the original manuscript. You can check the manuscript from page 2 line 52 to 58.

1.4 Reviewer Comments

Since the scanning pattern visited each spot three times, it cannot be said that the total input energy is three times the process’s linear energy density, 0.943 J/mm. So reporting 2.83 J/m in the abstract is wrong because it conveys the meaning that the incident energy per melt is that much. The end product from three times may help homogenization, but the microstructure will be very different from single scan with three times the energy.

Reply of the author: Thank you for your kind reminding. It is our unclear report and we are sorry about this. The energy density 2.83 J/mm is actually unreasonable as the energy powder absorbed. According to comment, we have made a specific demonstration and you can find the specific revises which has been highlighted in the page 3 line 96 to 98 in revision

1.5 Reviewer Comments

It is mentioned that three samples wer eus.

Reply of the author: We can’t understand what this comment means. We are very sorry for about this.

1.6 Reviewer Comments

It will be easier to read Figure 2 with elemental references embedded in the image. Also the third graph needs y-axis label to refer to the property being measured.

Reply of the author: Thank you very much for your correcting suggestion and we have already add the label to third graph in Figure 2. Meanwhile, we have checked the manuscript several times to avoid the similar mistakes.

1.7 Reviewer Comments

I can see some smaller peaks than those labelled in Figure 3’s XRD pattern. All peaks need to be labeled since here the technique is being used to conclude the single phase formation.

Reply of the author: Thank you very much for your constructive suggestion. In the revision we have add another pattern which contains the other BCC tungsten phase in the alloy and you can find the obvious difference between two pattern in Figure 3. Therefore, from the comparison, we confirm the alloy composition is uniform.

1.8 Reviewer Comments

Figure 5’s labels are a little hard to read. I would suggest increasing font-size.

Reply of the author: Thank you for your kind reminding. We have already increased label font-size into a suitable size. Meanwhile, we have checked the manuscript several times to avoid the similar mistakes

1.9 Reviewer Comments

The microstructure in Figure 6(c) appears to be lamellar/widmanstatten (like in Ti64 high cooling rate) rather than dendritic/snow-like as described by the authors. I suggest the authors reconsider their interpretation or provide other images that better capture/represent the real microstructure.

Reply of the author: Thank you very much for your constructive suggestion. According to your comments, we have confirmed that many lamellar martensite structure dendrites have appeared in the pattern and have further explained in the revised manuscript which has been highlighted in the page 8 Fig 6.

1.10 Reviewer Comments

Since SLM structures are anisotropic in nature with columnar grain structure that is their trademark, the microstructure of the XZ and XY planes should be presented.

Reply of the author: Thank you for your kind reminding. We have labeled the XY planes in the Figure 6 and have further instructed in revised manuscript.

1.11 Reviewer Comments

Similarly, it is also important to know the orientation of sample in Hardness testing. It will be good to compare the hardness data from XY and XZ planes.

Reply of the author: Thank you very much for your constructive suggestion. Due to the influence of process, we need to devote to more effort in printing thicker samples. Therefore, in this study, we really sorry for not studying the hardness of XZ plane. This part of the work will be carried out in future research

We have also modified the whole text in order to present our work more properly. Some typographical and grammatical errors have been removed and corrected. We hope that these revisions are satisfactory and that the revised version will be acceptable for publication in International journal of heat and mass transfer. If you have any question about this paper, please don’t hesitate to let me know.

Thank you very much for your work concerning our paper.

Wish you all the best!

Sincerely,

Prof. Wang

fuwang@xjtu.edu.cn

Reviewer 2 Report

In the present study, the authors have used selective laser melting (SLM) for manufacturing NbMoTaW refractory high entropy alloy (HEA). The investigated HEA is substitute material for superalloys. The authors have also highlighted several advantages of the SLM technique over the conventional manufacturing processes. The manuscript is well-written and deserves to be published. I have few queries and suggestions for the authors.

(1)In the introduction the authors have highlighted two key challenges for the refractory HEAs: low ductility and inhomogeneous distribution of elements. However, as mentioned by the authors none of these issues can be completely resolved by SLM process. So, the authors should define clearly why SLM can be advantageous. I suggest to add in the introduction some of the points related to the three advantages mentioned towards the end of the manuscript.

(2)In page 4, the authors have mentioned that mixing process helps to improve the absorption. Can the authors describe the mechanism behind this observation? Also, what are the factors which determine the absorption for pure elements?

(3) In sub-section 3.2, XRD results indicated uniform composition. However, for the EDS measurements, the authors found “Ta and W firstly precipitate ……., and Nb and Mo shows segregation …..”. Aren’t the XRD and EDS results contradictory?

(4)In page 6 it is mentioned that “Ta had the …. even up to 28.21%, Nb had the lowest ratio of 21.50% ….”. These values are only averaged values. For Ta, the value can go up to 29.49 and for Nb, the value can be as low as 20.97.

(5)The language needs to be checked at some places.

Author Response

Dear reviewer:

We are very grateful to your comments for the manuscript (ID materials-442714 " Manufacturing High-Performance Refractory High-entropy Alloy via Selective Laser Melting (SLM) "). Based on these comments and suggestions, we have made careful modifications on the original manuscript. All changes made to the text are highlighted in revision. Below you will find our point-by-point response to the reviewers’ comments/suggestions: 

1.     Specific Comments:

1.1 Reviewer Comments

In the introduction the authors have highlighted two key challenges for the refractory HEAs: low ductility and inhomogeneous distribution of elements. However, as mentioned by the authors none of these issues can be completely resolved by SLM process. So, the authors should define clearly why SLM can be advantageous. I suggest to add in the introduction some of the points related to the three advantages mentioned towards the end of the manuscript.

Reply of the author: Thanks for your kind suggestion. Compared with traditional manufacturing methods, SLM has the advantage that its molten pool can reach several thousand degrees instantaneously under the action of laser, and it has enough energy to melt and mix high melting point metals. Besides, because of the grain refinement caused by the rapid cooling process, the properties of the forming parts are also improved. Therefore, SLM has great potential for forming ultra-high melting point metals. According to your comments, we have made a special introduction in revised manuscript which has been highlighted in the page 2 line 62 to 65 in revision.

1.2 Reviewer Comments

In page 4, the authors have mentioned that mixing process helps to improve the absorption. Can the authors describe the mechanism behind this observation? Also, what are the factors which determine the absorption for pure elements?

Reply of the author: Thanks for your kind suggestion. We have also found this interesting phenomenon, but we have not done enough research on this issue before, so we can’t give a reasonable and accurate explanation. For pure elements, the factors of sphericity, powders size and laser wave length etc determine the laser absorption of powders. For this part content, we will gradually improve in the future research.

1.3 Reviewer Comments

In sub-section 3.2, XRD results indicated uniform composition. However, for the EDS measurements, the authors found “Ta and W firstly precipitate ……., and Nb and Mo shows segregation …..”. Aren’t the XRD and EDS results contradictory?

Reply of the author: Thank you very much for your reminding. Because XRD is a macroscopic test, when the content of precipitated phase is low, it may not show the diffraction peak. But EDS is a relatively small microscopic analysis. In this case, the proportion of precipitated phase is relatively large. Therefore, the precipitation can be detected. In summary, the XRD and EDS results are not contradictory actually.

1.4 Reviewer Comments

In page 6 it is mentioned that “Ta had the …. even up to 28.21%, Nb had the lowest ratio of 21.50% ….”. These values are only averaged values. For Ta, the value can go up to 29.49 and for Nb, the value can be as low as 20.97.

Reply of the author: Thanks for your kind reminding. According to your comments, we have explained the data more specifically in revised manuscript which has been highlighted in the page 6 line 158 to 160 in revision.

1.5 Reviewer Comments

The language needs to be checked at some places.

Reply of the author: Thanks for your kind reminding. According to your comments, we have revised some of the language in this article which has been highlighted in the page 8 from line 208 to 210, page 9 from line 248 to 250, page 10 from line 251 to 252 and page 10 from line 266 to 268 in revision.

We have also modified the whole text in order to present our work more properly. Some typographical and grammatical errors have been removed and corrected. We hope that these revisions are satisfactory and that the revised version will be acceptable for publication in International journal of heat and mass transfer. If you have any question about this paper, please don’t hesitate to let me know.

Thank you very much for your work concerning our paper.

Wish you all the best!

Sincerely,

Prof. Wang

fuwang@xjtu.edu.cn

Reviewer 3 Report

Following comments needs to be addressed:

Line 55 and 82: Some of the standardized steel numbering system needs to be used (EN, SAE, DIN, JIS) instead of 45#.

Line 67: In the chapter 2, the description of approach of microstructure evaluation is missing, e.g. in what depth the samples were sectioned and analysed, what was the built direction in relation to the microstructure cut, etc.

Line 83: The description of the size of the sample is misleading and needs to be clarified “10 mm×10 mm×0.5 mm (height, or unlimited height)“. What was the height of the samples used in the analyses?.

Was it really only 0.5 mm? In case that only the samples of height 0.5mm were fabricated, some conclusions seem to be too strong (e.g. lines 140 and 255), because processing itself is not managed. SLM processing of thin layers may be significantly different to processing of bulk material due to different cooling rate. Processing of bulk may result in cracking or overheating and high porosity, which may be an insurmountable problem.

Line 119: The chapter 3 Results and Discussion does not contain any density results of the fabricated part, which was mentioned on the line 110. Results needs to be added.

Line 202, Figure 6.: The orientation of the sample regarding the built direction needs to be added, e.g. built direction arrow into the pictures or the description in the figure caption.

Line 248: The arrangement of pictures in Figure 9. should be standard, (a) on the left side (b) on the right side, now it is opposite which is misleading when reading the figure caption.

Comments regarding language:

Line 110: Use of “drainage” is inappropriate term for naming the porosity evaluation method.

Line 204 and 205: The sentence about Hall-Petch law needs to be reformulated.

Lines 245 – 247: The sentence needs to be reformulated, especially the part “two kinds of patterns both corrosion strongly“.

Lines 248 – 249: The sentence needs to be reformulated, especially the part “the corrosion 248 deep can up to 1 mm“.

Line 256: The abbreviation et al should be replaced by etc.

Line 264: The sentence needs to be reformulated, the verb is missing in the first part.

Author Response

Dear reviewer:

We are very grateful to your comments for the manuscript (ID materials-442714 " Manufacturing High-Performance Refractory High-entropy Alloy via Selective Laser Melting (SLM) "). Based on these comments and suggestions, we have made careful modifications on the original manuscript. All changes made to the text are highlighted in revision. Below you will find our point-by-point response to the reviewers’ comments/suggestions: 

1.     Specific Comments:

1.1 Reviewer Comments

Line 55 and 82: Some of the standardized steel numbering system needs to be used (EN, SAE, DIN, JIS) instead of 45#.

Reply of the author: Thanks for your kind reminding. According to your comments, we have changed standardized steel number as C45 in revised manuscript which has been highlighted in the page 2 line 67 and page 3 line 86 in revision.

1.2 Reviewer Comments

Line 67: In the chapter 2, the description of approach of microstructure evaluation is missing, e.g. in what depth the samples were sectioned and analysed, what was the built direction in relation to the microstructure cut, etc.

Reply of the author: Thanks for your kind reminding. In this study, microstructure and hardness of the sample surfaces were analyzed on XY plane. According to your comments, we have made a specific instruction about this part in revised manuscript which has been highlighted in the page 3 line 102 to 104 in revision.

1.3 Reviewer Comments

Line 83: The description of the size of the sample is misleading and needs to be clarified “10 mm×10 mm×0.5 mm (height, or unlimited height)”. What was the height of the samples used in the analyses?

Reply of the author: Thanks for your kind reminding. In this study, the sample used to microstructure and hardness analysis are all with the size of 10 mm×10 mm×0.5 mm and the analytical sites are all on the surface of the samples. We have made a specific instruction about this part in revised manuscript which has been highlighted in the page 3 line 86 to 87 in revision.

1.4 Reviewer Comments

Was it really only 0.5 mm? In case that only the samples of height 0.5mm were fabricated, some conclusions seem to be too strong (e.g. lines 140 and 255), because processing itself is not managed. SLM processing of thin layers may be significantly different to processing of bulk material due to different cooling rate. Processing of bulk may result in cracking or overheating and high porosity, which may be an insurmountable problem.

Reply of the author: Thanks for your kind reminding. We are really sorry for this situation. The emphasis of this study is to show that SLM method has great potential for superalloy forming, and the samples formed by SLM method can have high properties. Due to the influence of process, we need to devote to more effort in printing thicker samples. This part of the content will be gradually improved in the future research.

1.5 Reviewer Comments

Line 119: The chapter 3 Results and Discussion does not contain any density results of the fabricated part, which was mentioned on the line 110. Results needs to be added.

Reply of the author: Thanks for your kind reminding. We are really sorry for this reckless mistakes. Actually, we had tested the density of pattern. But, because of the special size of the pattern, the measurement error of density by Archimedes method is large, so we preliminarily infer that the pattern has high density from the surface gloss. We have made a specific instruction about this part in revised manuscript which has been highlighted in the page 5 line 136 to 137 in revision.

1.6 Reviewer Comments

Line 202, Figure 6.: The orientation of the sample regarding the built direction needs to be added, e.g. built direction arrow into the pictures or the description in the figure caption.

Reply of the author: Thanks for your kind suggestion. We are really sorry for this negligence. In Fig 6, we have added the label regarding the built direction in revision.

1.7 Reviewer Comments

Line 248: The arrangement of pictures in Figure 9. should be standard, (a) on the left side (b) on the right side, now it is opposite which is misleading when reading the figure caption.

Reply of the author: Thanks for your kind reminding. We are really sorry for this reckless mistake. We have modified the picture arrangement as your wish.

1.8 Reviewer Comments

Line 110: Use of “drainage” is inappropriate term for naming the porosity evaluation method.

Reply of the author: Thanks for your kind reminding. As response in comment 5,

we had tested the density of pattern. But, because of the special size of the pattern, the measurement error of density by Archimedes method is large, so we preliminarily infer that the pattern has high density from the surface gloss. We have made a specific instruction about this part in revised manuscript which has been highlighted in the page 5 line 136 to 137 in revision.

1.9 Reviewer Comments

Line 204 and 205: The sentence about Hall-Petch law needs to be reformulated.

Reply of the author: Thanks for your kind reminding. We have reformulated the sentence in revised manuscript which has been highlighted in page 8 from line 208 to 210.

1.10 Reviewer Comments

Lines 245 – 247: The sentence needs to be reformulated, especially the part “two kinds of patterns both corrosion strongly”.

Reply of the author: Thanks for your kind reminding. We have reformulated the sentence in revised manuscript which has been highlighted in page 9 from line 248 to 250.

1.11 Reviewer Comments

Lines 248 – 249: The sentence needs to be reformulated, especially the part “the corrosion 248 deep can up to 1 mm”.

Reply of the author: Thanks for your kind reminding. We have reformulated the sentence in revised manuscript which has been highlighted in page 10 from line 251 to 252.

1.12 Reviewer Comments

       Line 256: The abbreviation et al should be replaced by etc.

Reply of the author: Thanks for your kind reminding. We have replaced all “et al” to “etc” in the revised manuscript.

1.13 Reviewer Comments

Line 264: The sentence needs to be reformulated, the verb is missing in the first part.

Reply of the author: Thanks for your kind reminding. We have reformulated the sentence in revised manuscript which has been highlighted in page 10 from line 266 to 268.

We have also modified the whole text in order to present our work more properly. Some typographical and grammatical errors have been removed and corrected. We hope that these revisions are satisfactory and that the revised version will be acceptable for publication in International journal of heat and mass transfer. If you have any question about this paper, please don’t hesitate to let me know.

Thank you very much for your work concerning our paper.

Wish you all the best!

Sincerely,

Prof. Wang

fuwang@xjtu.edu.cn

Reviewer 4 Report

It is interested that 4 different powders were mixed for HEA SLM use in this study. It focuses on composition, microstructure and corrosion properties. 

1.     The topic should focus on the main idea of this study

2.     please identified clearly the challenges of NbTaMoW HEA alloys using casting and other traditional process in the introduction part.

3.     In sentence, what is the plasma ? SLM process using laser.

4.     It did not mention powder size and flowability of each element before mixing in material and method. It should provide the results for mixed powders.

5.     Each layer 3 times melting during the SLM Process. What is parameters of each times for melting and SLM parameters should be provided (such as powder, scan speed, build direction, scan strategic and hatch distance

6.     There is not clear of scale bar, number in figure 2. There are some porosity and gas bubbles in each powders. It did not shown the absorption of mix powder and distribution/morphology of mixing powders.

7.     XRD shows only BCC of NbMoTaW. It did not find the other phase. It must be mentioned about that. Also the evidence of the densification of SLMed should be provided.

8.     There is martensite microstructure common seen in SLM process because high cooling process. It never found it in this study and only fine dendrite. Have you found any martensite in this study

Author Response

Dear reviewer:

We are very grateful to your comments for the manuscript (ID materials-442714 " Manufacturing High-Performance Refractory High-entropy Alloy via Selective Laser Melting (SLM) "). Based on these comments and suggestions, we have made careful modifications on the original manuscript. All changes made to the text are highlighted in revision. Below you will find our point-by-point response to the reviewers’ comments/suggestions: 

1.     Specific Comments:

1.1 Reviewer Comments

The topic should focus on the main idea of this study.

Reply of the author: Thank you very much for your kind suggestion. In this study, laser sintering of NbMoTaW alloy was carried out by SLM method, and the properties and structure of the samples were analyzed. The topic has been modified in revision.

1.2 Reviewer Comments

Please identified clearly the challenges of NbMoTaW HEA alloys using casting and other traditional process in the introduction part.

Reply of the author: Thank you very much for your kind suggestion. Traditional process for super alloy is casting or arc melting. For casting process, the highest melting temperature is only 1800°C, which is far from enough for the NbMoTaW alloy in this study. Besides, although the arc melting can up to the melting point, this method can't form arbitrary shape. We have made a specific instruction about this part in revised manuscript which has been highlighted in the page 2 line 45 to 49 in revision.

1.3 Reviewer Comments

In sentence, what is the plasma? SLM process using laser.

Reply of the author: Thank you very much for your query. The “plasma” means plasma emission. It's part of the thermal process of laser sintering instead of the heat resource. We have made a more clearly instruction about this part in revised manuscript which has been highlighted in the page 2 line 53 to 55 in revision.

1.4 Reviewer Comments

It did not mention powder size and flowability of each element before mixing in material and method. It should provide the results for mixed powders.

Reply of the author: Thank you very much for your kind suggestion. We did not test the particle fluidity, but the flour spreading process was good. Besides, the content about this part can been seen as the result of experimental powders and we have improved the data in this section which has been highlighted in the page 4 line 128 to 129 in revision.

1.5 Reviewer Comments

Each layer 3 times melting during the SLM Process. What is parameters of each times for melting and SLM parameters should be provided (such as powder, scan speed, build direction, scan strategic and hatch distance.

Reply of the author: Thanks for your professional suggestion. In this study, the three times melting parameters for each layer are the same. The original parameters were chosen as follows: hatch distance ds was 0.1 mm, dz was 0.1 mm, fp was 0.8 and v was 250 mm/s, with S-cross scanning in x and y direction and we have add the parameter hatch distance in revised manuscript which has been highlighted in the page 3 line 88 to 93 in revision.

1.6 Reviewer Comments

There is not clear of scale bar, number in figure 2. There are some porosity and gas bubbles in each powders. It did not show the absorption of mix powder and distribution/morphology of mixing powders.

Reply of the author: Thanks for your kind reminding. We are really sorry for this negligence and have modified in revised manuscript which has been highlighted in the figure 2.

1.7 Reviewer Comments

XRD shows only BCC of NbMoTaW. It did not find the other phase. It must be mentioned about that. Also the evidence of the densification of SLMed should be provided.

Reply of the author: Thanks for your kind reminding. We have mentioned that the macroscopic composition is relatively uniform in the pattern. Besides, because of the special size of the pattern, the measurement error of density by drainage method is large, so we preliminarily infer that the pattern has high density from the surface gloss. We have made a more clearly instruction about this part in revised manuscript which has been highlighted in the page 5 line 136 to 137 in revision.

1.8 Reviewer Comments

There is martensite microstructure common seen in SLM process because high cooling process. It never found it in this study and only fine dendrite. Have you found any martensite in this study.

Reply of the author: Thanks for your professional suggestion. According to your comments, we have confirmed that many lamellar martensite structure dendrites have appeared in the pattern and have further explained in the revised manuscript which has been highlighted in the page 7 line 193 to 195.

We have also modified the whole text in order to present our work more properly. Some typographical and grammatical errors have been removed and corrected. We hope that these revisions are satisfactory and that the revised version will be acceptable for publication in International journal of heat and mass transfer. If you have any question about this paper, please don’t hesitate to let me know.

Thank you very much for your work concerning our paper.

Wish you all the best!

Sincerely,

Prof. Wang

fuwang@xjtu.edu.cn

Round  2

Reviewer 1 Report

I am glad to see the significant improvement in the manuscript. It would have been a stronger paper if mechanical properties are included but I understand limitations from sample material. I think the manuscript is publication ready now.Good luck!